# Role of Agricultural Management in the Provision of Ecosystem Services in Warm Climate Vineyards: Functional Prediction of Genes Involved in Nutrient Cycling and Carbon Sequestration

**DOI:** 10.3390/plants12030527

**Published:** 2023-01-23

**Authors:** Rafael Alcalá-Herrera, Beatriz Moreno, Martin Aguirrebengoa, Silvia Winter, Ana Belén Robles-Cruz, María Eugenia Ramos-Font, Emilio Benítez

**Affiliations:** 1Department of Biotechnology and Environmental Protection, Estación Experimental del Zaidín, CSIC, c/Profesor Albareda 1, 18008 Granada, Spain; 2Department of Crop Sciences, Institute of Plant Protection, University of Natural Resources and Life Sciences, Gregor-Mendel-Straße 33, 1180 Vienna, Austria; 3Assessment, Restoration and Protection of Mediterranean Agrosystems Service (SERPAM), Estación Experimental del Zaidín, CSIC, c/Profesor Albareda, 1, 18008 Granada, Spain

**Keywords:** vineyard, cover vegetation, ecosystem functions, nutrient cycling, soil bacteria

## Abstract

(1) Background: Maintaining soil fertility and crop productivity using natural microbial diversity could be a feasible approach for achieving sustainable development in agriculture. In this study, we compared soils from vineyards under organic and conventional management by predicting functional profiles through metagenomic analysis based on the 16S rRNA gene. (2) Methods: The structure, diversity and predictive functions of soil bacteria related to the biogeochemical cycle of the soil were analyzed, including oxidative and hydrolytic C-cycling enzymes, N-cycling enzymes and P-cycling enzymes. The inter-row spontaneous vegetation in the organic vineyards was also characterized. (3) Results: A clear effect of the farming system (organic vs. conventional) and cover management (herbicides plus tillage, mowing only and mowing plus tillage) on bacterial beta diversity and predicted functions was evidenced. While conventional viticulture increased the potential capacity of the soil to regulate the cycling of inorganic forms of N, organic viticulture in general enhanced those functions involving organic N, P and C substrates. Although the soil bacterial community responded differently to contrasting soil management strategies, nutrient cycling and carbon sequestration functions remained preserved, suggesting a high bacterial functional redundancy in the soil in any case. However, most of the predicted bacterial functions related to soil organic matter turnover were enhanced by organic management. (4) Conclusions: We posit the potential for organic viticulture to adequately address climate change adaptation in the context of sustainable agriculture.

## 1. Introduction

The growing need for sustainable food production for the worldwide human population requires balancing the demand for food production, maintaining good environmental conditions and protecting biodiversity. In the last century, intensive production has led to a sharp increase in productivity due to increasing use of pesticides, fertilizers, irrigation and heavy machinery, with associated negative effects, such as landscape simplification, biodiversity decline and environmental degradation [1,2].

Land-use change to more sustainable practices in the so-called biodiversity-based systems has been proposed as a feasible alternative to increase the overall level of ecosystem service provision, such as recycling of nutrients, producing foods, regulating pests, storing carbon and maintaining the soil structure [3]. However, the high degree of unpredictability, the variability of particular biodiversity-based practices, such as organic farming, and site-specific responses may hamper in many cases the viability of organic management for ecosystem service provision [4].

Viticulture and winemaking are important socioeconomic sectors [5], with Italy, France and Spain being the three main wine producers worldwide [6]. The optimal temperature ranges for grape cultivation are delimited by a lower threshold necessary for fruit ripening and an upper threshold that would lead to damaged fruit in addition to certain minimum requirements for soil moisture during the growing season [7]. Consequently, rising temperatures and changed rainfall patterns due to climate change may have severely negative effects for many regional economies. The potential of organic farming to mitigate the influence of agriculture on global warming has been widely reviewed [8,9]. One of the main challenges for mitigating and adapting to climate change in viticulture, and for organic agriculture in general, relates to understanding how management practices affect the soil microbiota that drive soil functions, such as biogeochemical cycling and carbon sequestration [10].

Microorganisms are crucial for maintaining soil quality for sustainable plant growth and their role in the release of mineral nutrients through the decomposition of organic matter or the recycling of minerals has been evidenced [11]. From this perspective, the efficiency in the recycling of nutrients from organic compounds for decomposers has been proposed as the key parameter that controls ecosystem processes [12]. In this scenario, maintaining soil fertility and crop productivity using natural microbial diversity could be the best approach to achieve the Sustainability Agreements in the agri-food supply chain foreseen by the new EU Common Agricultural Policy for the period 2023–2027 [13] and the European Commission’s Farm to Fork strategy target of increasing organic farming to 25% and halving pesticide use by 2030 [14].

The importance of soil microbial communities for ecosystem services is also related to the positive effect of diversity on multifunctionality, i.e., the simultaneous maintenance of multiple ecosystem functions [12,15]. Despite the great diversity of microorganisms in the soil, most biogeochemical transformations seem to be mediated by a limited set of metabolic pathways present in a variety of taxonomic groups [16]. Furthermore, the functional redundancy of soil microbial communities is thought to be responsible for maintaining the stability over time of the associated ecosystem function [17,18]. Relationships between functional and taxonomic diversity using metagenomic-based approaches have proven to be a feasible tool to study the prediction of microbial biogeochemical processes, as well as to identify the extent to which microbial functional redundancy regulates dynamic ecological flows [19]. The evidence grows but research in permanent crops, such as, vineyards is especially scarce.

A huge number of studies acknowledge the contribution of the soil microbiome to soil quality and health. In a recent review, more than 40 functions of the soil microbiome that promote the health of soil, plants, animals, and humans have been identified [20]. However, effects on microbial communities are often controversial when organic and conventional systems are compared, finding positive, negative, or no effects of soil management on microbial diversity [21,22], which could be related to the large differences in respect to soil management within the different farming systems. Furthermore, the effects depend on the landscape heterogeneity and on organism groups studied [23]. It has also been reported that soil microbial communities do not necessarily differ between conventional and organic agriculture [24]. In this context, it is necessary to understand to what extent different agricultural soil management practices, such as tillage, herbicide application and/or mowing, modify the soil microbiome.

In this study, we hypothesized that the management system history affects the bacterial contribution to soil ecosystem services in vineyards. We investigated the relationship between soil management in organic and conventional cultivation and the abundance of bacterial genes that encode key enzymes of the C, N, and P cycles obtained by 16S functional prediction.

## 2. Results

### 2.1. Cover Vegetation

Due to the intensive inter-row management, conventional vineyards were not covered by any vegetation. Regarding the inter-row vegetation of the organic farms, the O3 organic vineyards with mowing cover management showed the highest species richness (10.5 species m^−2^) and diversity (H’= 0.5 bits). The species composition was dominated by Geraniaceae (*Erodium* sp. and *Geranium molle*), followed by Asteraceae and Poaceae (Appendix A). The rest of the organic farms showed lower species richness (range: 4–7) and diversity (range: 0.005 to 0.08) and were clearly dominated by *Brassicaceae,* mostly *Diplotaxis* sp. Plant cover ranked from 57.5% to 14.4% without a clear pattern related to the type of cover management.

### 2.2. Soil Bacteria Community Structure and Diversity

More than half of the detected phyla were Proteobacteria and Actinobacteria, with remaining phyla attributed to 10 other phyla (Figure 1). Significant differences between management systems (conventional vs. organic) were attributed to minor phyla Firmicutes (*p*-value: 0.004, F-value: 11.72) and Nitrospirota (*p*-value: 0.0158; F-value: 7.53). At the class level, differences between management systems were imputed to classes Bacilli (*p*-value: 0.0004, F-value: 21.55), Acidimicrobiia (*p*-value: 0.047, F-value: 4.74), Nitrospiria (*p*-value: 0.015, F-value: 7.53), Rubrobacteria (*p*-value: 0.020, F-value: 6.86), the unclassified_Actinobacteriota (*p*-value: 0.002, F-value: 14.22) and Chloroflexi (*p*-value: 0.077, F-value: 3.60). However, no differences in alpha diversity (Chao1) at the ASV level (*p*-value: 0.176; [*t*-test] statistic: 1.46) could be attributed to management.

On closer analysis, we detected some differences at the family level, and the LefSe analysis showed that four families, Micrococcaceae, Azospirillaceae, Oxalobacteraceae and Rubrobacteriaceae were mainly responsible for the differences between managements (Figure 2A). At the genus level, *Massilia*, *Arthrobacter* and *Rubrobacter* showed an interesting abundance distribution pattern following, from higher to lower abundance, the gradient: herbicide + tillage > mowing only > mowing + tillage (Figure 2B). *Sphingomonas*, *Microvirga* and unclassified_Actinobacteriota were higher under organic farming with mowing as cover crop management.

Figure 3 shows the differences in species complexity, that is, beta diversity, between different farming systems and vegetation cover managements of vineyard soils. A clear farming system effect (*p*-value: 0.001, F-value: 2.35) was evidenced, as well as a cover management effect (*p*-value: 0.001, F-value: 2.93) on the Bray-Curtis dissimilarity based on abundance (Figure 3A). Interestingly, cover management separates two different groups within organic management, linked to mowing only and mowing plus tillage (Figure 3B). No effect on bacterial community structure was attributed to row or interrow spatial location (*p*-value: 0.915, F-value: 0.61).

### 2.3. Predictive Metagenomic Profiles

A marked increase in the nitrogen metabolic pathways, both in the reduction (assimilatory and dissimilatory nitrate reduction, denitrification and nitrogen fixation) and oxidation pathways (nitrification), was observed under conventional management (Figure 4). On the contrary, organic management enhanced the potential ammonification pathway in the soils.

Organic farming increased the potential functional molecular pathways controlled by phosphatase enzymes (Figure 5). Both acid and alkaline phosphatases catalyze the reaction: phosphate monoester + H_2_O = alcohol + phosphate, but both differ in the optimal pH of catalysis, 6 in the former and above 7.5 in the latter.

The bacterial production potential of hydrolase and oxidoreductase enzymes related to the biogeochemical carbon cycle is shown in Figure 6 and Figure 7, respectively. Organic management generally boosted the potential synthesis of C-degrading hydrolytic enzymes, with the exception of the hydrolases mannosyl-glycoprotein endo-beta-N-acetylglucosaminidase (NAG) and 3-beta-D-glucan glucohydrolase (β-Glucanase) (Figure 6).

Oxidoreductase enzymes, which carry out both synthetic and degradative reactions, followed a different pattern (Figure 7). While the potential production of oxygenases, such as phenol 2-monooxygenase (*pmo* NADPH), *pmo* NADH and toluene monooxygenase system protein (*tmo*), and the catalase-peroxidase (*katG*) was higher under organic management, conventional management increased the potential synthesis of NADH peroxidase (*npr*) and 2,4-dichlorophenol 6-monooxygenase (*tfdB*) oxygenases in vineyard soils.

Looking at the deduced genes related to bacterial atmospheric carbon fixation, we found that the KEGG module related to global Calvin-Benson-Bassham cycle and specifically bacteria harboring the ribulose-bisphosphate carboxylase gene (*cbbL*) encoding the enzyme ribulose-1,5-bisphosphate carboxylase-oxygenase (RuBisCO) were favored by conventional management (Figure 8).

Non-metric multidimensional scaling (NMDS) analysis separately showed the effect of cover management type on the predicted genes involved in C, N, and P cycling and C sequestration in conventional and organic vineyards (Figure 9). Mowing had the opposite effect of herbicide application and tillage, which similarly affected potential bacterial functions.

## 3. Discussion

There is a common agreement on the need to maintain stable ecosystem functions associated with soil microorganisms in agricultural systems. In this study, we report the effect of agricultural management on soil bacterial community and its predicted functions in organic and conventional vineyards.

No important differences in bacterial composition were found between the main bacterial phyla but between the lower taxa across the two farming systems. Previous research also shows minor compositional changes due to different farming systems and fertilization styles, thus evidencing a stable microbial community in the soil [25]. In this sense, many studies propose that organic farming facilitates a copiotrophic lifestyle compared to conventional management [26]. However, many other studies show the opposite [27], attributing in any case the effect to the greater quantity/availability of nutrients. In our study, although no differences were found in soil nutrient content between systems, we did find an increase in the copiotrophic Firmicutes and Nitrospiriia under conventional management. However, it could be wrong to apply a specific lifestyle classification to all members of a phylum since not all members of a taxonomic group share the same ecological characteristics. In this context, it has been shown that management responses could occur at lower taxonomic levels [28]. On closer analysis, differences between the two vineyard cultivation systems could be attributed to families and genera ascribed to Actinobacteria (*Arthrobacter* and *Rubrobacter*) and Beta- (*Massilia*) and Alpha-Proteobacteria (*Sphingomonas* and *Microvirga*). Conventional management had a positive effect on the first two, but the ascription of these (sub)phyla to a certain trophic lifestyle still remains controversial. Although some studies associate Actinobacteria and Beta-proteobacteria with copyotrophy, others suggest that these phyla are better adapted to oligotrophic conditions [29]. The same inconsistency in the ascription occurs for Alpha-Proteobacteria, enhanced in our study by organic agriculture.

No differences in alpha diversity (diversity within a farming system) were attributed to management. In contrast, differences were found in species complexity between farming systems (beta diversity) at the ASV level. As expected, either the presence or absence of spontaneous vegetation, herbicide use plus tillage or tillage plus mowing, or the type of fertilization influenced the structure of the bacterial population. Previous studies have indicated that the beta diversity of plant-associated bacteria is strongly related to the beta diversity of plants estimated by both species and biomass composition [30]. Despite the fact that the spontaneous vegetation did not follow a pattern of abundance and diversity within the organic systems, in our study an effect of cover management on bacterial diversity was detected. We suggest that both cover management (mowing or mowing plus tillage) and quantity/quality or the organic amendments drove the change in bacterial species complexity in organic vineyards.

It is increasingly recognized that functional diversity patterns can provide more useful information on land use impacts than taxonomic richness [31,32]. To test multifunctionality, we considered some key bacterial functions related to soil biogeochemical cycling, including oxidative and hydrolytic C-cycling enzymes, N-cycling enzymes and P-cycling enzymes.

There are a large number of studies that point to the role of soil management practices on the abundance and expression of functional genes involved in the soil nitrogen cycle [33]. Here, we report the abundance of predicted functional genes of bacteria involved in five steps of soil N cycling (nitrification, denitrification, nitrate reduction, nitrogen fixation and ammonification) in vineyards under two different management systems. Contrary to expectations, conventional vineyard management increased the abundance of most of the predicted genes involved in soil nitrogen metabolism. There is general agreement on the positive effect of organic fertilization on nitrogen cycling genes, commonly associated with higher inputs of labile C and N from organic amendments and reduced tillage [34]. Thus, compared to conventional systems, organic management usually increases the abundance and activity of nitrate reduction, nitrification, denitrification and nitrogen fixation genes [35,36]. Despite this consensus, some controversy still exists. Some research shows that organic fertilization alone is not enough to stimulate nitrate-reducing activity [37,38]. Other studies also reveal an organic management effect only on *nosZ* abundance and no effect on the other denitrification genes, such as *nirK* and *nrfA*, or those involved in fixation and nitrification, such as *nifH* and *amoA*, respectively [39]. Furthermore, inorganic chemical fertilizers can enhance the abundance of the bacterial nitrification *amoA* gene when nitrogen is available in semi-arid conditions [40]. Finally, there is general agreement on the effect of vegetation cover on denitrification. Since the process takes place under low-oxygen or purely anoxic conditions, the demonstrated potential of cover vegetation to improve soil structure and reduce compaction, and therefore improve aeration, could be related to the decrease in soil denitrification potential detected in organic vineyards.

There is also strong evidence for the relationship between nitrogen fixation and sustainable agriculture [41], including lower numbers of *nif* genes and free-living nitrogen-fixing bacteria under inorganic N fertilization and no tillage [42,43,44]. In the present study, organic inputs from both cover vegetation and organic amendments did not improve the number of predicted genes involved in N fixation. This is consistent with research results showing that conventional agriculture increased overall nitrogen fixation activity in bulk soils [45,46].

Conversely, organic management improved potential ammonification in the soil, assessed as the predicted abundance of glutamate dehydrogenase genes (GDH2, *gudB*, *rocG*, GLUD1_2, *gdhA*). Since ammonification involves the mineralization of low molecular weight organic molecules, it is feasible to relate it to the greater availability of organic substrates derived from organic amendments and plant residues in this agricultural system.

Considering agricultural practices within conventional vineyards, we must also take into account the effects of herbicides on the rhizospheric microbial community. In this regard, the effect of glyphosate is still unclear. Some studies show that the herbicide affects the expression of N cycle genes, although in very high concentrations never used in agriculture [47,48]. Complementary studies show contradictory effects on the genes of the bacterial C cycle [48,49].

The bacterial genes *phoA,B,C* and *phoN*/PHO encode alkaline and acid phosphatase, respectively, enzymes involved in the hydrolysis of soil organic phosphorus into phosphate. It is well known that functional groups of bacteria associated with phosphorus cycling are highly sensitive to agricultural management practices [50]. In our study, the abundance of predicted phosphatase genes was significantly higher under organic management. It is well known that soil phosphatases respond to organic P but not mineral P, even long-term P inputs reduce not only abundance but diversity of phosphatase genes [51]. Overall, the relative abundance of predicted alkaline phosphatase genes was dominant over predicted acid phosphatase genes, something expected in neutral or alkaline soils. Interestingly, alkaline phosphatase-encoding bacterial species in organically managed soils have been shown to be more closely related to acid phosphatase-encoding species than in conventional management [52]. It has also been evidenced that, under organic management, the predominant short-term effect on phosphorus-mineralizing bacterial communities in acid soils can be exclusively attributed to a rhizosphere effect, whereas *phoD*-harboring bacteria are influenced by both organic matter addition as by the soil rhizosphere [53]. In our study, only in organic vineyards with spontaneous vegetation cover could the above trend be verified, since conventional soils were always bare of vegetation. In this context, it is widely accepted that cover crops generally increase microbial P biomass and consequently phosphatase activity and also the diversity and species richness of *phoA,B,D*-harboring bacteria [54]. Furthermore, cover vegetation tends to be highly effective in systems with little available P, as is the case with our experimental system, due to their ability to access “unavailable” P reserves [55].

Focusing on tillage management, there is controversy about the effect on functional genes associated with the P cycle. There is evidence that tillage tends to increase the abundance of functional genes associated with the degradation of P compounds [56]. Conversely, soil management practices that minimize soil disturbance have been shown to improve phosphorus availability and therefore soil phosphatase activities [57]. In accordance with the latter, we found that the deduced functional profiles associated with the P cycle were favored by organic management that included spontaneous vegetation cover and zero or minimum tillage.

It has been established that the long-term continuous supply of organic fertilizers results in an increase in hydrolytic enzymes that degrade carbon compounds in the soil [58]. In addition, there is a general agreement on the positive effect of the management of cover crops and zero tillage in the C cycle [59,60]. There is less consensus on the effect of inorganic fertilization on the genes associated with C degradation in the soil [61]. Some research suggests that long-term inorganic fertilization accelerates soil C turnover in agroecosystems, with concomitant upregulation of most genes involved in C degradation [62], while others show the opposite [63].

In our study, organic vineyard management overall boosted the abundance of deduced bacterial genes involved in the C cycle. For this study, we considered separately those genes that strictly produce hydrolases from those that produce oxidoreductase enzymes. The latter are involved not only in C mineralization but also in polycondensation reactions in the soil from aromatic monomers, such as phenol or catechol, therefore they play a key role in the stabilization of organic carbon in the soil.

We detected in organic vineyards an increase in the predicted bacterial genes involved in the hydrolysis of the most abundant polysaccharides, such as glucose, xylan and cellobiose, with the exception of the enzyme that catalyzes the hydrolysis of beta-D-glucose to alpha-glucose. We also found a detrimental effect of organic management on the predicted genes encoding the enzyme N-acetyl-glucosaminidase (NAG). NAG is responsible for microbial N acquisition through chitin degradation and has often been considered as an N-related enzyme [64]. The increase under conventional management somewhat followed the same trend found in the most important soil enzymes involved in the N cycle.

With respect to oxidoreductases, the organic system increased the abundance of the deduced genes encoding the enzymes phenol and toluene monooxygenase, capable of converting benzene to phenol and catechol. The same was true for genes that produce catalase-peroxidase, an oxidoreductase enzyme that is thought to break down H_2_O_2_ and is often correlated with heterotrophic decomposition and humification [65]. In contrast, conventional management was attributed to an increase in the predicted abundance of *npr* and *tfdB* genes. The former involves the production of dichlorocatechol from dichlorophenol. The latter is a stricto sensu peroxidase with a suggested role in both lignin degradation and carbon humification [66].

There is an increase in the demand for knowledge about the concept of carbon stability in soil, directly related to the sustainability of ecosystem services. Changes in soil organic carbon stability can alter soil carbon release and consequently atmospheric CO_2_ concentration [67]. In this context, carbon accumulation and sequestration has been pointed out as a critical factor. We studied the effect of management on the bacterial potential to fix atmospheric carbon. The Calvin cycle is the predominant pathway for bacteria to assimilate CO_2_ and is therefore an important process of the soil carbon cycle. RuBisCO enzymes catalyze the carbon dioxide assimilation reaction, the first rate-limiting step of the Calvin cycle. Contrary to available information [68], conventional vineyard management enhanced the predicted abundance of *cbbL* genes encoding the enzyme RuBisCO, thus increasing the CO_2_ assimilation potential of autotrophic soil bacteria.

However, despite the importance of soil carbon sequestration in the global carbon cycle, carbon fixation does not directly imply carbon stability and multiple factors affect the soil C turnover, particularly temperature in a global warming scenario. In this context, the stability of C has been associated with the process of humification of organic matter in the soil. A recent study also shows that humic acids in the soil enhance heat stress tolerance to plants [69]. We found in the organic management a general increase of deduced genes that code for oxidoreductase enzymes, mainly of the oxygenase type. While those of the oxidases type are fundamentally involved in energy metabolism, oxygenases are attributed both a synthetic and a degrading role. Although there is much controversy about how humification of organic matter in soil occurs, oxidative coupling of phenols has been recognized as a key step in the polyphenol humification theory, where catechol and o-quinones play a fundamental role in the synthesis of humic substances [70]. In addition, it is well known that the increase in catechol also promotes the formation of humic substances in abiotic reactions in the catechol-Maillard system [71]. However, in no case did we find lignin-degrading genes recognized for their role in humification according to the lignin theory [72], except for very low amounts of dye-decolorizing peroxidases [EC 1.11.1.19].

## 4. Materials and Methods

### 4.1. Field Experiment and Sampling Site

The vineyards are located in the Designation of Origin Montilla-Moriles (Córdoba, Spain). The climate in this area is classified as semi-continental Mediterranean, with short winters and long, dry and hot summers (CsA according to Köppen climate classification [73]), with an average monthly temperature of 16.6 °C and an average accumulated rainfall of 54.6 mm [74]. Traditional sweet wines made from the Pedro Ximénez grape variety characterize this wine region.

The study was carried out between September 2020 and January 2021 on vine plants of the native cultivar Pedro Ximénez grafted onto Richter 110 rootstocks. Eight vineyards, four organic (O) and four conventional (C), over 20 years old under organic or conventional management, were selected from a larger project (https://www.biodiversa.eu/2022/10/31/secbivit/). Conventional vineyard fertilization is based on the application of NPK fertilizers one to three times a year in all vineyards, except in C1 where manure and NPK fertilizers alternate annually. On average, each NPK fertilizer application involves the addition of approximately 22 kg of ammonia, 16 kg of urea, 44 kg of P_2_O_5_ and 16 kg of K_2_O per hectare. Spontaneous vegetation is removed by tillage and a 36% glyphosate herbicide is applied 1.5 times per year, resulting in permanently bare soil. The organic vineyards are fertilized with vermicompost tea (O1 and O2) or manure (O3 and O4) once a year. For O4, a commercial organic fertilizer is also applied once or twice a year. Spontaneous vegetation is managed by mowing plus tillage (O1 and O4) or mowing alone (O2 and O3). Regarding the control of fungal diseases, growers annually applied sulfur at a rate of 12 kg ha^−1^ in O1 and O2, zero treatment in O3 and copper and sulfur oxychloride at a rate of 6.8 and 9.45 kg ha^−1^, respectively, in O4. In conventional vineyards, 27.6 kg ha^−1^ of sulfur were applied in C1, 25 kg ha^−1^ of cuprocalcium sulfate in C2 and 50 kg ha^−1^ of sulfur in C3 and C4, plus two applications of synthetic fungicides at a rate of 2 kg ha^−1^ in the last.

Soil samples were collected in autumn from the 0–20 cm layer of both the rows and inter-rows. Each soil sample was an integrated sample made up of four subsamples taken from four different points separated by 3 m. Samples were gently sieved to pass through a mesh sieve of 2-mm, kept at 4 °C during the sampling campaign and then stored at −80 °C until DNA extractions.

### 4.2. Soil Characterization

According to the IUSS Working Group WRB [75], sandy loam soils of the treatments C1, C3, C4, O1, O2 and O3 are classified as Calcic Cambisols, C2-clay loam soil is classified as Vertic Cambisol and sandy loam soil of the O4 treatment as Calcaric Fluvisol.

Air-dried field soil samples were used to determine physicochemical properties at the Scientific Instrumentation Service, EEZ-CSIC, Granada, Spain (Appendix A). Soil pH was measured using a suspension of 1:2.5 soil:water with a pH-meter CyberScan PCD 6500 (Eutech Instruments, Nijkerk, The Netherlands). Total N and soil organic C were determined with the aid of the Leco-TruSpec CN elemental analyzer (LECO Corp., St Joseph, MI, USA). Total mineral content was determined by the digestion method with HNO_3_ 65%:HCl 35% (1:3; *v*-*v*) followed by analysis using inductively coupled plasma optical emission spectrometry (ICP-OES) (ICP 720-ES, Agilent, Santa Clara, CA, USA).

No differences were detected in the physicochemical characteristics of the soils between the two management systems.

### 4.3. Vegetation Sampling

The percentage of spontaneous vegetation in the field was estimated in mid-January 2021, one to two months before the farmers removed the vegetation cover. The estimation method was based on four 1m^2^ plots per vineyard, located at the center of the row separated by 6 m. Each plot was photographed and the different species and cover percentage were identified, based on an analysis of the pictures and species, and were named according to the Flora Vascular de Andalucía Oriental [76]. For each plot, the following parameters were estimated: (1) plant cover (percentage of soil covered by plants), (2) cover percentage per species (percentage of soil covered by species *I* or genera/family when species were unidentifiable), (3) cover percentage per family (percentage of soil covered by family *x*), (4) species richness (number of species) and (5) diversity (Shannon index), estimated as:(1)$$H^{\prime }=\overset{i=n}{\underset{i=1}{\sum }}p_{i}\;{Lnp}_{i}$$
where *p_i_* is the relative frequency of species *i* and *n* is the total number of species.

### 4.4. Molecular Analyses of Soil Bacteria

DNA was extracted from four 1 g aliquots of each soil sample using the bead-beating method with the aid of a PowerSoil^®^ DNA Isolation Kit (MoBio Laboratories, Solana Beach, CA, USA), following the manufacturer’s instructions. Extractions from the same subsample were pooled and concentrated at 35 °C to a final volume of 50 µL using a Savant Speedvac^®^ concentrator (Savant, Thermo Scientific, Holbrook, NY, USA).

The V3–V4 hypervariable regions of the 16S rRNA gene were targeted by bacterial PCR primers 5′ CCTACGGGNBGCASCAG 3′ and 5′ GACTACNVGGGTATCTAATCC 3′ [77,78] to characterize bacterial communities from two replicates per sample using the Illumina MiSeq (Illumina Inc., San Diego, CA, USA) (2 × 250 nucleotides paired-end protocol) at the genomic facilities of the López-Neyra Institute of Parasitology and Biomedicine (IPBLN-CSIC). The dada2 v1.24.0 pipeline [79] was used to process raw sequences and construct an amplicon sequence variant (ASV) table. ASV taxonomic assignment was achieved by implementing the *assignTaxonomy* function (based on naive Bayesian classifier method) against the SILVA v138.1 database [80]. An ASV × matrix was generated using the Marker Data Profiling module on the MicrobiomeAnalyst web platform (https://www.microbiomeanalyst.ca/faces/home.xhtml) [81,82]. All samples reached a plateau based on the rarefaction curves generated by the MicrobiomeAnalyst tool.

### 4.5. Predictive Metagenomic Profiles

Tax4fun v0.3.1 [83], implemented in MicrobiomeAnalyst’s Shotgun Data Profiling module, was used to predict, from 16S data sets obtained from the SILVAngs web server, the functional pathways of soil bacterial communities based on the Kyoto Encyclopedia of Genes and Genomes (KEGG) annotations [84]. These were based on (1) modules, that is, functional units of sets of genes in the KEGG metabolic pathways database that can be linked to specific metabolic capacities and other phenotypic characteristics and (2) KEGG Orthologous (KO) corresponding to a group of orthologous genes identified by a K number that have identical functions. Appendix A shows the modules and KOs focused on in this study.

### 4.6. Statistical and Diversity Analyses

Univariate general linear mixed models (GLMMs) were performed to assess the effect of the management system (conventional vs. organic) on the abundance of bacterial taxa and KEGG modules/KOs. Since two composite samples (row and interrow) were collected per vineyard, vineyard identity was included as a random factor in the GLMMs using *lme4* package [85] of R v4.2.2 statistical software [86]. The normality of the data and the homogeneity of the variance were evaluated with the Shapiro-Wilk test and the Levene statistic, respectively. The ASV abundance information was normalized to the abundance value of the sample with the least number of sequences.

Chao1 index of alpha diversity and beta diversity/linear discriminant effect size (LEfSe) analyses were performed with the *phyloseq* package to test for differences in species complexity within a sample and between groups, respectively, using the MicrobiomeAnalyst web platform. For beta diversity, Bray-Curtis distance and permutational ANOVA (PERMANOVA) were used to assess the distance between samples and the statistical significance of the clustering pattern, respectively. For LEfSe, features with an LDA score > 4 were considered important biomarkers of each treatment, and a *p* value < 0.05 indicates significant differences. Finally, non-metric multidimensional scaling (NMDS) analysis with Bray-Curtis distance was performed with the *vegan* package [87] to test dissimilarities in bacterial KEGG KOs due to management.

## 5. Conclusions

In this study, we evidenced that agricultural management history selected for bacteria with different potential to control nutrient cycles in the soil. Although the soil bacterial community responded differently to the contrasting soil management strategies, the nutrient cycling and carbon sequestration functions remained preserved, suggesting a high bacterial functional redundancy in the soil, which positively affects the stability of the microbial community and resilience to disturbance. Redundancy refers to the ability of a microorganism to perform one function at the same rate as another under the same environmental conditions, and we found that most of the bacterial functions related to organic matter turnover could potentially develop at a higher rate in organic viticulture. Since there is a positive relationship between humification and plant environmental stress tolerance, we postulate the potential of organic viticulture to adequately address climate change adaptation in the context of sustainable agriculture. However, more empirical work, including landscape and location dependency studies, is needed.

## Figures and Tables

**Figure 1 plants-12-00527-f001:**
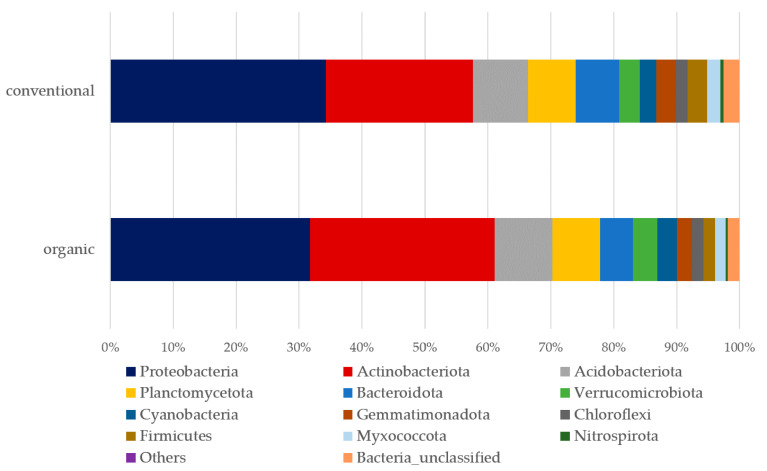
Relative abundance of soil bacteria in conventional and organic vineyards.

**Figure 2 plants-12-00527-f002:**
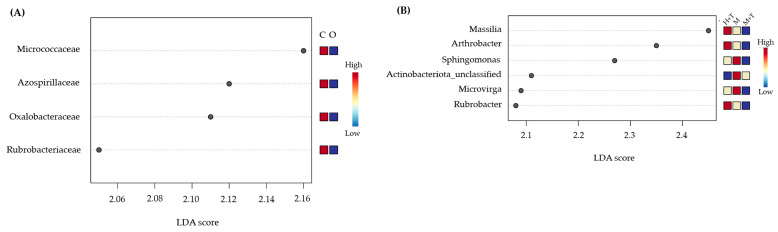
Linear discriminant analysis (LDA) scores and heatmap from blue (low) via white (medium) to red (high) of relative abundances in (**A**) conventional (C) and organic (O) systems and, (**B**) herbicide + tillage (H + T), mowing only (M) and mowing + tillage (M + T) cover managements.

**Figure 3 plants-12-00527-f003:**
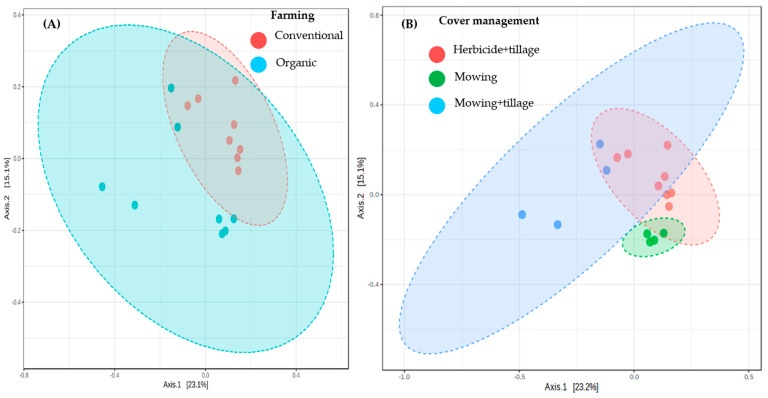
Effect of farming (**A**) and cover management (**B**) on bacterial beta diversity in vineyards’ soils. Ordination method PCoA; distance method: Bray-Curtis index; statistical method: PERMANOVA. Ellipses mean that 95% of the data fell inside the ellipse.

**Figure 4 plants-12-00527-f004:**
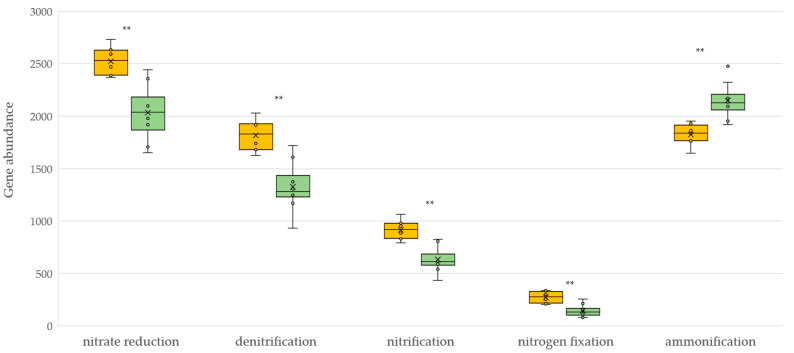
KEGG pathway molecular functions of nitrogen cycle for soils under conventional (yellow) and organic (green) management. Asterisk represents a significant difference (** *p* < 0.01).

**Figure 5 plants-12-00527-f005:**
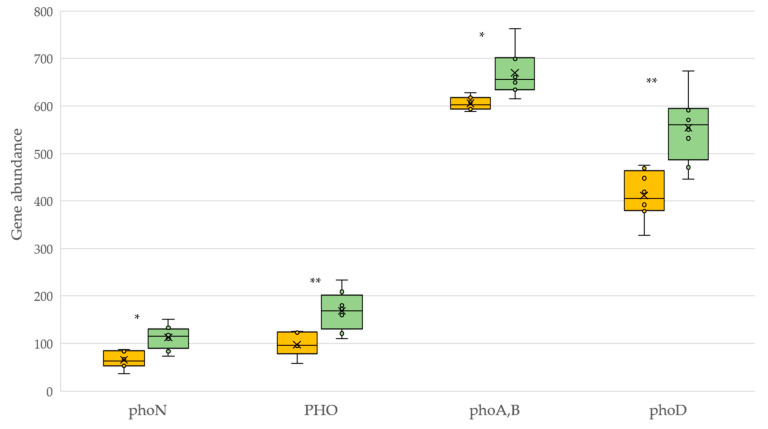
KEGG pathway molecular functions of organic phosphorus cycle for soils under conventional (yellow) and organic (green) management. *phoN*: acid phosphatase (class A) [EC 3.1.3.2]; PHO: acid phosphatase [EC 3.1.3.2]; *phoA,B*: alkaline phosphatase [EC 3.1.3.1]; *phoD*: alkaline phosphatase [EC 3.1.3.1]. Asterisk represents a significant difference (** *p* < 0.01, * *p* < 0.05).

**Figure 6 plants-12-00527-f006:**
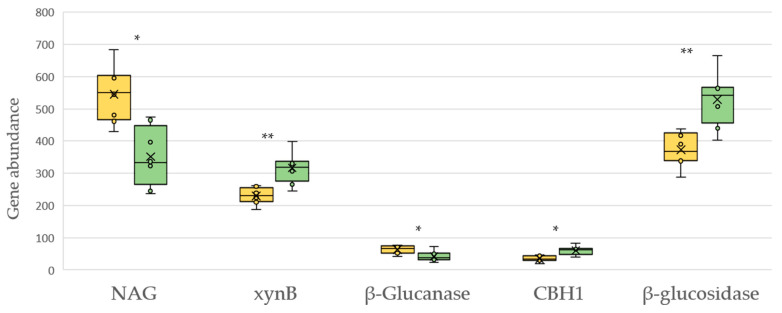
KEGG molecular functions of genes encoding carbon hydrolases in soils under conventional (yellow) and organic (green) management. NAG: mannosyl-glycoprotein endo-beta-N-acetylglucosaminidase [EC 3.2.1.96]; *xynB*: xylan 1,4-beta-xylosidase [EC 3.2.1.37]; β-Glucanase: 3-beta-D-glucan glucohydrolase [EC 3.2.1.58]; CBH1: cellulose 1,4-beta-cellobiosidase [EC 3.2.1.91]; β-glucosidase [EC 3.2.1.21]. Asterisk represents a significant difference (** *p* < 0.01, * *p* < 0.05).

**Figure 7 plants-12-00527-f007:**
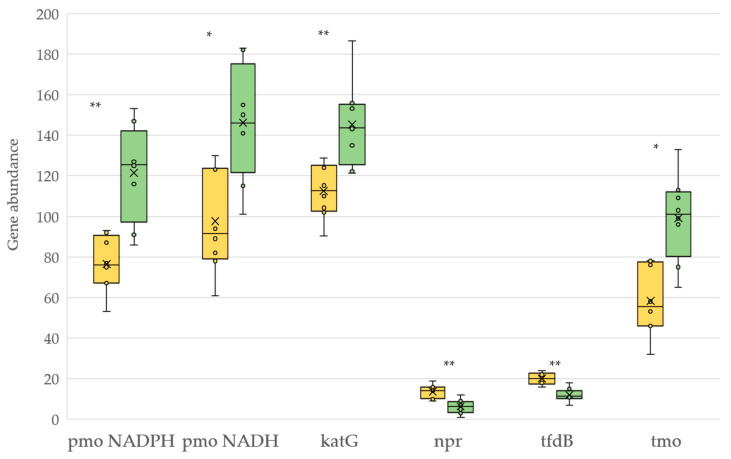
KEGG pathway molecular functions of carbon oxidoreductases for soils under conventional (yellow) and organic (green) management. *pmo* NADPH: phenol 2-monooxygenase (NADPH) [EC 1.14.13.7]; *pmo* NADH: phenol 2-monooxygenase (NADH) [EC 1.14.13.244]; *katG*: catalase-peroxidase [EC 1.11.1.21]; *npr*: NADH peroxidase [EC 1.11.1.1]; *tfdB*: 2,4-dichlorophenol 6-monooxygenase [EC 1.14.13.20]; *tmo*: toluene monooxygenase system protein [EC 1.14.13.236]. Asterisk represents a significant difference (** *p* < 0.01, * *p* < 0.05).

**Figure 8 plants-12-00527-f008:**
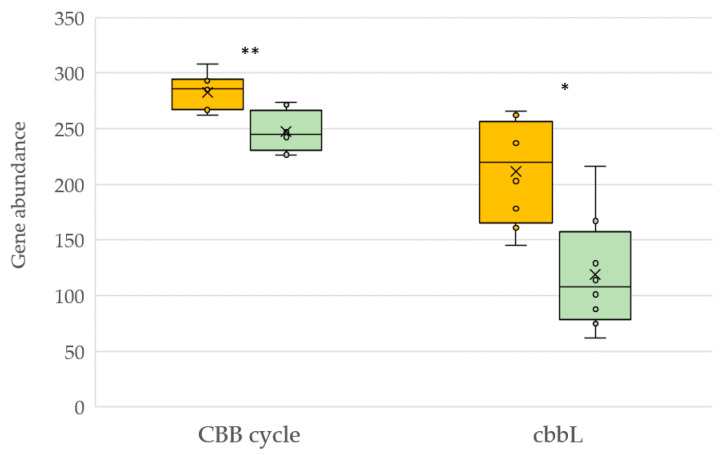
KEGG pathway molecular functions of carbon fixation for soils under conventional (yellow) and organic (green) management. CBB: Calvin-Benson-Bassham cycle; *cbbL*: ribulose-bisphosphate carboxylase [EC 4.1.1.39]. Asterisk represents a significant difference (** *p* < 0.01, * *p* < 0.05).

**Figure 9 plants-12-00527-f009:**
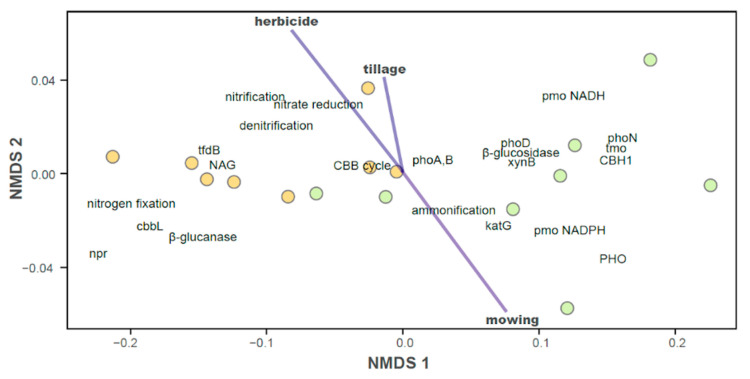
NMDS plot showing mean position of each bacterial KEGG KO according to cover management practices (herbicide, tillage, mowing) as dissimilarity vectors. Each point represents a vineyard sample: conventional (yellow) and organic (green) management. Vector significance: herbicide R^2^ = 0.636, *p*-value = 0.001; tillage R^2^ = 0.430, *p*-value = 0.018; mowing R^2^ = 0.636, *p*-value = 0.001. Management (conventional, organic) significance: R^2^ = 0.577, *p*-value = 0.001. Stress = 0.006.

## Data Availability

All raw Illumina sequence data were deposited in the Sequence Read Archive (SRA) service of the NCBI database (https://www.ncbi.nlm.nih.gov/) (BioProject ID: PRJNA907731).

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
