# Peer review of "Role of Agricultural Management in the Provision of Ecosystem Services in Warm Climate Vineyards: Functional Prediction of Genes Involved in Nutrient Cycling and Carbon Sequestration"

_plants, 2023, doi:10.3390/plants12030527_

Round 1

Reviewer 1 Report

This work provides scientific evidence of interest to promote an alternative to conventional soil management. Here are some comments that could be considered to improve the manuscript:

-         - This work includes a study of the vegetation and it is not reflected in the abstract provided.

-          - Linea 51: A bibliographic citation should be included for this confirmation: “in addition to certain minimum requirements for soil moisture during the growing season”.

-          - Lines 93-97: In the planted hypothesis, it should be specified that they study the relationship between soil management in conventional and organic cultivation. There are other differentiating elements between both farming systems that are not studied in this work.

-        -  It would be interesting to include in the materials section the type of soil where each of the studied vineyards are planted.

Author Response

This work provides scientific evidence of interest to promote an alternative to conventional soil management. Here are some comments that could be considered to improve the manuscript:

Thank you for the perceptive and constructive review of the manuscript Plants-2144879. The manuscript has been substantially improved by your comments and suggestions.

 - This work includes a study of the vegetation and it is not reflected in the abstract provided.

Answer: A statement on vegetation is now included in the abstract.

- Line 51: A bibliographic citation should be included for this confirmation: “in addition to certain minimum requirements for soil moisture during the growing season”.

Answer: The reviewer is right. A new reference has been included that supports whether this statement or the previous one (line 52).

Lines 93-97: In the planted hypothesis, it should be specified that they study the relationship between soil management in conventional and organic cultivation. There are other differentiating elements between both farming systems that are not studied in this work.

Answer: We agree, changed (lines 96-97).

It would be interesting to include in the materials section the type of soil where each of the studied vineyards are planted.

Answer: The type of soil and, following the suggestion of reviewer 2, some physical parameters, have been included (lines 420-422).

Reviewer 2 Report

The paper is very interesting and rich of data, but some modifications/improvements are needed, as follows.

Lines 99-108: please double check the data, because  they do not seem consistent with  Table S1.

Figure 1: please enlarge the legend.

Lines 222- 224: because, most likely there is a higher  organic C level in the conventional farming soil than in the organic farming soil.

Lines 306-310:… that’s why it is important to know the soil pH of the experimental vineyard.

Line 386: when was the experiment performed? The monthly meteorological conditions of that year have to be indicated  (temperature, rainfall).

Line 392: are the vines grown in  the 8 vineyards grafted on the same rootstock? If yes, which one? It is actually known that the plant genotype (the root system) affects the composition and activity  of the soil microbiome.

Lines 394-395: which is the mean annual supply of N, P and K (kg/ha)?

Line 397: what is the meaning of “1.5 times per year”?

Line 398: is it possible to estimate  the mean annual supply of N. P, K (kg/ha)?

Line 401: since the experiment compares conventional vs. organic farming, it is crucial to know how much copper/ha/year (if any)  was utilized for spray treatments, even though  downy mildew pression, in that environment, is supposed to be very low; moreover it is crucial to know how much sulfur/ha/year (if any) was utilized for spray treatments against powdery mildew.

Line 402: when was the soil sampled?

Line 407: Soil pH should be provided. No data on the physical properties (i.e. texture)?

Lines 409-410: how do you explain the lower  organic C level in conventional farming soil  as compared to organic farming soil (Table S2)?

Line 417: when was the vegetation sampling done?

Author Response

The paper is very interesting and rich of data, but some modifications/improvements are needed, as follows.

Thank you for the perceptive and constructive review of the manuscript Plants-2144879. The manuscript has been substantially improved by your comments and suggestions.

Lines 99-108: please double check the data, because they do not seem consistent with Table S1.

Answer: We apologize for the mistake. It is already corrected (lines 102-107).

Figure 1: please enlarge the legend.

Answer: The legend has been enlarged (line 124).

Lines 222- 224: because, most likely there is a higher organic C level in the conventional farming soil than in the organic farming soil.

Answer: This is indeed a feasible reason. However, in our study no significant differences in total organic carbon were detected due to management (ANOVA F= 0.5767, p= 0.4602). There is a comment about it in lines 431-432.

Lines 306-310:… that’s why it is important to know the soil pH of the experimental vineyard.

Answer: Totally agree. Now pH values have been included in Table S2.

Line 386: when was the experiment performed? The monthly meteorological conditions of that year have to be indicated (temperature, rainfall).

Answer: Information and references are now included in lines 392-395 following the reviewer’s suggestion.

Line 392: are the vines grown in the 8 vineyards grafted on the same rootstock? If yes, which one? It is actually known that the plant genotype (the root system) affects the composition and activity of the soil microbiome.

Answer: We totally agree and sorry for forgetting to mention it. Winegrowers in the area prefer the autochthonous variety Pedro Ximénez grafted onto Richter 110 rootstocks. Information is now included in line 396.

Lines 394-395: which is the mean annual supply of N, P and K (kg/ha)?

Answer:  In general, winegrowers applied 547 kg/ ha of a 7-5-12 NPK fertilizer; that is, around 22 kg of ammonia, 16 kg of urea, 44 kg of P2O5 and 16 kg of K2O per hectare.  Now included in lines 401-403.

Line 397: what is the meaning of “1.5 times per year”?

Answer: We realize that this can be confusing. We would like to point out that some years the growers apply once and others twice, but unfortunately there is not a rule for this. Considering the last few years, the average number of application result in 1.5.

Line 398: is it possible to estimate the mean annual supply of N. P, K (kg/ha)?

Answer: I'm afraid it's almost impossible. Organic farmers in the area apply manure or sometimes vermicompost but the composition depends on availability or on starting substrates, sometimes the manure comes from cows, sometimes from sheep. We realize it's a data interpretation issue, but it's the way farmers can handle it.

The thing is that they have worked like this for at least 20 years before our experiment, and it could be assumed that the soil composition is stable at least in each of the farms.

Line 401: since the experiment compares conventional vs. organic farming, it is crucial to know how much copper/ha/year (if any) was utilized for spray treatments, even though downy mildew pression, in that environment, is supposed to be very low; moreover, it is crucial to know how much sulfur/ha/year (if any) was utilized for spray treatments against powdery mildew.

Answer: We have included how growers control fungal diseases. Even if downy mildew pressure is really low, every farm in the area is different, i.e. there is a range from zero to synthetic fungicide application (lines 408-413)

Line 402: when was the soil sampled?

Answer: Included in the M&M section (line 414) following the reviewer’s suggestion.

Line 407: Soil pH should be provided. No data on the physical properties (i.e. texture)?

Answer: Texture, pH and soil classification have been included in point 4.2 Soil Characterization (line 419) following the reviewer’s suggestion.

Lines 409-410: how do you explain the lower  organic C level in conventional farming soil  as compared to organic farming soil (Table S2)?

Answer: no significant differences in total organic carbon were detected due to management (ANOVA F= 0.5767, p= 0.4602).  You can find a comment about it in lines 431-432.

Line 417: when was the vegetation sampling done?

Answer: Included in the M&M section (line 435) following the reviewer’s suggestion.